# The impact of sport engagement on life satisfaction, mental and psychological well-being among athletes

Arif Özsarı[1]*, Halil Uysal[2], Gültekin Lekesiz[1], Mehmet Çağrı Çetin[1], Murat Tilki[2], Erkan Gülgösteren[1], Tolga Tek[3], Mehmet Altın[3]

**1** Faculty of Sport Sciences, Mersin University, Mersin, Turkey, **2** Institute of Education Sciences, Mersin University, Mersin, Turkey, **3** Faculty of Sport Sciences, Selcuk University, Konya, Turkey

* arifozsari@mersin.edu.tr

**Citation:** Özsarı A, Uysal H, Lekesiz G, Çetin MÇ, Tilki M, Gülgösteren E, et al. (2026) The impact of sport engagement on life satisfaction, mental and psychological well-being among athletes. PLoS One 21(1): e0340069. https://doi.org/10.1371/journal.pone.0340069

## Abstract

Physical inactivity is a prevalent issue worldwide with concerning implications for public health. Regular participation in sporting activities is associated with numerous physical, psychological, and social benefits. This research aimed to examine the the impact of sport engagement on sportive life satisfaction, mental well-being, and psychological well-being among 473 Turkish athletes, comprising 246 females and 227 males, from various sports disciplines: football, rugby, wrestling, kickboxing, archery, table tennis, taekwondo, volleyball, swimming, cycling, handball, and karate. Research data were collected through validated scales measuring sport engagement, sportive life satisfaction, mental well-being, and psychological well-being. In addition to descriptive statistics, hypotheses formulated within the research model were tested via correlation and multiple regression analyses within the relational model framework. Correlation analysis demonstrated a positive association between sport engagement and all three well-being dimensions. Multiple regression analyses further confirmed that sport engagement significantly and positively predicted sportive life satisfaction, mental and psychological well-being. Sports participation should be encouraged to enhance life satisfaction, mental and psychological well-being.

## Introduction

Although physical activity has been proven to play a key role in promoting overall health and well-being, inadequate physical activity in individuals has become a global problem [1]. In addition, the World Health Organization (WHO) and many other scientific studies emphasize the benefits of physical activity on general health [2–5,6,7]. In addition to its physical benefits, sport has many mental and psychological benefits. It can be stated that participation in sports as a part of social life in societies varies in many ways such as entertainment, pleasure, physical development, healthy living and socialization. Participation in sports is a positive experience especially for

**Data availability statement:** All relevant data are within the paper and its Supporting Information files.

**Funding:** The author(s) received no specific funding for this work.

**Competing interests:** The authors have declared that no competing interests exist.

children and young people and should be encouraged for all individuals [8–10]. Regular participation in sports also plays an important role for adults to adopt a physically and mentally healthy lifestyle [11,12].

It is emphasized that the sense of engagement is an important factor in individuals' attitudes towards the activities (such as sports) they participate in [13]. Engagement is expressed as the level of closeness to the phenomenon that the individual feels positive emotions [14] and it is also stated that engagement emphasizes a positive situation [15]. Considering that the sense of engagement is an important factor in sports participation, it is stated that engagement in sports and physical activities is the consistent and continuous reflection of emotions such as sportive effort, enjoyment, feeling fit and belief in the activity and the mood of the individual [16]. According to Guillen and Martinez-Alvarado [17], the concept of sport engagement is explained by three basic values: fitness, dedication and internalization. Fitness can be explained by the fact that athletes are always mentally strong; dedication is the individual's sense of responsibility and fulfillment of their duties with a high level of motivation; and internalization can be explained by the athletes' immersion in their duties.

In order for individuals to feel happy both in the society they live in and individually, their subjective well-being should follow a positive trend. One of the basic components of subjective well-being is life satisfaction [18]. Life satisfaction is a cognitive process that is closely linked to an individual's self-assessment of quality of life based on personal standards and mental health indicators such as anxiety and depression that affect overall health and well-being [19]. According to Fleming [20], life satisfaction is the state of being individually satisfied with one's life. It is expressed as a phenomenon that can shed light not only on a specific situation or event experienced by the individual, but also on the experiences of the individual's whole life in general [21]. In other words, life satisfaction refers to the judgment processes by evaluating the life of the individual with a holistic approach [22,23]. One can argue that life satisfaction contributes positively to an individual's ability to resist the difficulties experienced in life [24]. Poor life satisfaction is associated with deterioration of mental and physical health. Physical activity is also a major factor that positively affects life satisfaction [25]. On the other hand, sportive life satisfaction is the application of the concept of life satisfaction to sport. From this point of view, sportive life satisfaction can be defined as evaluating all aspects of sport in a positive way, obtaining pleasure, happiness and satisfaction from these aspects, and perceiving the conditions, expectations, needs, wishes and desires of sporting life within the framework of the criteria determined by the individual [26].

According to WHO [27] mental well-being is a state of mind that enables people to cope with the stresses of life, realize their abilities, learn and work well, and contribute to society. This concept can also be expressed as being emotionally, cognitively and environmentally satisfied with one's own behavior and not feeling psychological fluctuations [28]. The notion of mental well-being is often used together or interchangeably with the concept of mental health [29,30]. Therefore, we can suggest that mental well-being is a protective factor in important aspects such as health and illness [31]. There is compelling evidence for the effects of sport on mental health and mental well-being [32].

Psychological well-being, on the other hand, refers to the values that an individual possesses and the ability to use these values effectively in order to cope with difficult situations, strengthen oneself, acquire and develop new skills [33]. Psychological well-being, which individuals accept as a reflection of a good life, is an important aspect of social health [7]. According to Ryff [34], it refers to the development and self-realization of the individual by living in a satisfied way with one's life. Psychological well-being is also said to consist of such basic components as having a positive attitude about oneself and one's past, maintaining positive relationships with other people, one's autonomy, one's environmental awareness, one's life purpose, and a sense of feeling that one has developed their potential [35].

Today, negative effects of a physically passive lifestyle can be seen not only on an individual level but also in societal terms. As a matter of fact, health expenditures lead countries to face huge financial costs from a financial perspective [36]. WHO reports that one in four adults and four in five adolescents globally do not fulfill the requirements for participation in physical activity and that physical inactivity is one of the main causes of the increase in obesity rates and deaths from noncommunicable diseases, but also emphasizes that participation in physical activities recommended by WHO could prevent the loss of approximately 4–5 million lives annually [5]. Without a positive change in the physical inactivity attitudes of individuals worldwide, approximately 499 million new cases of non-communicable diseases caused by physical inactivity may occur by 2030 and approximately 215 million (43%) of these cases are expected to result from psychological and mental disorders [37].

Participation in regular physical activity has been reported to provide various physical and psychological health benefits [38–43]. Physical activity is particularly linked to improvements in the health and welfare levels of societies, especially when regular participation is ensured [44]. It goes without saying that a physically and mentally healthy society can be created as well as the socialization of individuals as long as sporting activities are practiced regularly by adhering to the sports culture throughout the society. If regular athletic activities are considered a key factor to ensure significant enhancements in individuals' life satisfaction levels, mental and psychological well-being levels, it can be regarded as a treasure that should be exploited to increase the welfare of communities and accordingly to create productive, efficient and strong societies in terms of communication. This study, which was conducted with athletes participating in various sports branches, aimed to examine the relationship between sportive life satisfaction, mental and psychological well-being levels of individuals participating in sports regularly and the effects of sports at individual and social level. In this context, the importance of this study is increasing by the emphasis placed on paticipation in physical activity in scientific studies, as well as the clear demonstration in reports published by the WHO [5] that a significant portion of health problems can be eliminated through paticipation in physical activity. This study, which examines life satisfaction and mental and psychological well-being levels, paticularly among active athletes, highlights the benefits of physical activity for individuals and society. It is anticipated that this study is essential for encouraging individuals to participate in physical avtivities. This study, which allows for a multifaceted assessment by considering the variables of sports engagement, sportive life satisfaction, mental and psychological well-being, which have been considered separately in previous studies, is considered original.

## Method

### Research Model

The general survey model is a screening design carried out on the whole population or a group or sample from the population in order to make a general judgment about the entire population consisting of diverse elements. Relational surveys can be conducted with these models. Relational survey models are research models that aim to determine the presence or degree of covariance between variables [45]. In this study, the relational survey model was used. The reason for preferring the relational survey model is that, instead of looking for a cause-effect relationship between variables, the variables included in the model are more likely to have a common effect on individuals' participation in physical activity and the changes that can be observed afterward. The hypotheses created within the scope of the research model are presented below.

H1: Vigor, which is a subdimension of sport engagement, has a positive effect on the sportive life satisfaction.

H2: Dedication, a subdimension of sport engagement, influences sportive life satisfaction positively.

H3: Vigor, a subdimension of sport engagement, will positively influence mental well-being.

H4: Dedication, a subdimension of sport engagement, has a positive effect on mental well-being.

H5: Vigor, a subdimension of sport engagement, has a positive effect on psychological well-being.

H6: Dedication, a subdimension of sport engagement, influences psychological well-being positively.

The hpyotheses of this study were tested with the variables and sample group selected in accordance with the relational survey model. It is thought that with the change in the sports engagement variable and its subdimensions, changes can also be seen in individuals' sportive life satisfaction, mental and psychological well-being levels. Hypotheses were created from variables that have high potential to represent motivation to participate in physical activities and positive psychological changes that can be seen afterwards. The flow chart of the research and analysis processes is presented below (Fig 1).

## Ethics committee approval information

To ensure ethical conduct, this research underwent a thorough review and received approval from the Scientific Research and Publication Ethics Committee of Osmaniye Korkut Ata University (document number 2023/1/1, dated 29/03/2023). The study adhered to the research principles of the Declaration of Helsinki. Informed consent forms were obtained from all Participants.

## Participants

In total, 473 athletes, 246 women (52%) and 227 men (48%), selected by convenience sampling method, participated in the study. In convenience sampling technique, researchers attempt to collect data from the easiest and most accessible individuals until they reach the sample size needed for the study [46]. The size or adequacy of sample size for varying population levels can be calculated through various statistical formulas [47]. In quantitatively oriented research, minimum acceptable sample sizes are employed for different populations at the 95% reliability level. Considering the number of research participants ($n = 473$), this sample size has the capacity to represent a population of ten million [46]. On the other hand, there is a possibility of bias in the convenience sampling method because the sample selection can be determined by the researcher. Due to the lack of randomization, the convenience sampling method has questionable aspects in terms of generalizability. Due to the earthquake in Turkey, the data was collected between July and September 2024. Although ethical approval for the study had previously been obtained, the suspension of sporting activities in the country due to the earthquake caused a delay in data collection. Because it was anticipated that the participating athletes would experience some social and psychological trauma due to the earthquake, data collection was delayed until sporting activities resumed and the athletes returned to routine physical activities. Given that the traumatic environment in which the athletes were

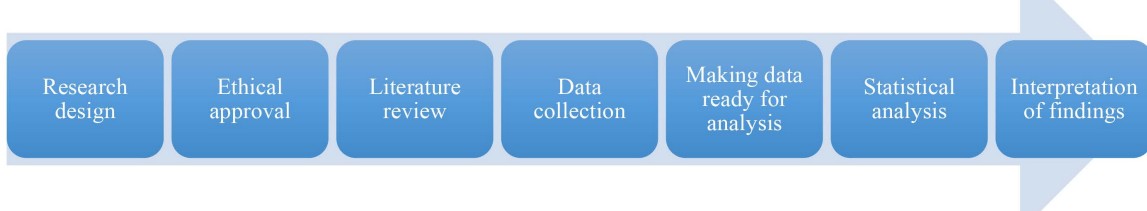

**Fig 1. Research and analysis flow chart.**

located could be misleading, the data collection period was extended, and the study was conducted with athletes who were able to return to active sports. No information that could identify individual participants was accessed during or after data collection.

The findings regarding the demographic characteristics of the participants are presented in Table 1. A total of 473 athletes participated in this study, with a relatively balanced gender distribution (52% female, 48% male). The athletes represented a variety of sporting disciplines, with the most common being taekwondo (20.9%, n = 99), archery (19%, n = 90), rugby (13.1%, n = 62), kickboxing (8.9%, n = 42), wrestling (8.2%, n = 39), volleyball (8%, n = 38), football (6.8%, n = 32), table tennis (4.7%, n = 22). The remaining participants represented smaller proportions of athletes from swimming (2.3%, n = 11), cycling (3%, n = 14), handball (2.3%, n = 11), and karate (2.7%, n = 13).

## Instruments

**Sport Engagement Scale (SES).** Developed by Guillen & Martinez-Alvarado [17] and adapted to Turkish by Kayhan et al. [48], the SES measures sport engagement through two subdimensions: vigor and determination. This 10-item scale utilizes a 7-point Likert-type response format. During the adaptation process of the scale to Turkish, the opinions of 6 expert and correlation analysis were used and it was seen that the Turkish scale had language equivalence. To examine the construct validity of scale, EFA and CFA analyses were conducted with independent sample groups. KMO (0.94) and Barlett' test of sphericity ($\chi2 = 3330,65$, p < 0,001) showed that the structure was suitable for discovery. It was observed that the sub-dimensions of the scale explained 67.59% of the total variance. Within the scope of CFA analysis, the fit indices of the scale (GFI = .95, TLI = .96, IFI = .97, RMSEA = .07 etc.) were examined and it was observed that they were within the acceptable fit range. The reliability of the scale was examined using Cronbach' Alpha coefficients and Test-Re-Test (r = 0.94) method. The Cronbach α coefficient of the whole scale was calculated as.91; Cronbach α coefficients of sub-scales are.91 and.77.

**Sportive Life Satisfaction Scale (SLSS).** Originally developed by Diener et al. [49] to assess general life satisfaction, this scale was adapted for the sports context by Mangan [50] and subsequently translated into Turkish by Somoglu & Cihan [26]. The SLSS comprises 5 items, employs a 7-point Likert-type response style, and measures overall satisfaction with one's sporting life. In order to test the validity of the scale, content validity index (CVI), EFA and CFA analysis

**Table 1. Demographic information of the participants.**

| Gender | | N | % |
|---|---|---|---|
| | Female | 246 | 52 |
| | Male | 227 | 48 |
| Sport discipline | Football | 32 | 6.8 |
| | Rugby | 62 | 13.1 |
| | Wrestling | 39 | 8.2 |
| | Kickboxing | 42 | 8.9 |
| | Archery | 90 | 19 |
| | Table tennis | 22 | 4.7 |
| | Taekwondo | 99 | 20.9 |
| | Volleyball | 38 | 8 |
| | Swimming | 11 | 2.3 |
| | Cycling | 14 | 3 |
| | Handball | 11 | 2.3 |
| | Karate | 13 | 2.7 |
| | Total | 473 | 100 |

were examined. As a result of content validity analysis, the scale was found to represent the construct (CVI = 0.96). Exploratory factor analysis showed that a one-factor structure explains 81.730% of the total variance. As a result of CFA (RMSEA = .057, SRMR = .011, GFI = .98, NFI = .99, RFI = .98, CFI = .99, IFI = .99), it was observed that the fit indices were within acceptable ranges. The values obtained from the results of Cronbach's Alpha (0.87) and Test-Re-Test (r = 0.89) reliability analyzes showed that the scale is a reliable tool.

**Warwick-Edinburgh Mental Well-being Scale (WEMWBS).** The WEMWBS, developed by Tennant et al. [30] and adapted into Turkish by Keldal [51], is a 14 item scale assessing mental well-being through 5-point Likert responses. EFA and CFA analysis were conducted in independent sample groups to prove the validity of the scale. The scale was found to be suitable for factor analysis according to the results of KMO (.85) and Barlett' test of sphericity ($\chi2 = 2110,179$, p < 0,001). As a result of the analysis, it was found that 51% of the total variance was explained and the scale was unidimensional as in the original form. With CFA analysis, fit indices ($\chi2$/sd = 3.71, NFI = .94, RFI = .93, IFI = .96, CFI = .96, NNFI = .95, RMR = .054) were examined and it was observed that they were within the acceptable fit range. Cronbach's Alpha internal consistency reliability coefficient of the scale was found to be 0.92.

**Psychological Well-Being Scale (PWBS).** Developed by Diener et al. [52] and adapted into Turkish by Telef [53], the PWBS is an 8-item scale measuring psychological well-being on a single dimension. It employs a 7-point Likert-type response design. The scale was found to be suitable for factor analysis according to the results of KMO (.84) and Barlett' test of sphericity ($\chi2 = 1007,633$, p < 0,01). As a result of the exploratory factor analysis, it was found that the total explained variance was 42% and that the items were grouped under one factor. With CFA analysis, fit indices ($\chi2$/sd = 4,645, NFI = .94, RFI = .92, IFI = .95, CFI = .95, RMR = .04, RMSEA = .08) were examined and it was observed that they were within the acceptable fit range. The reliability study indicated that the Cronbach alpha coefficient was.80.

## Statistical analysis

Prior to the main analyses, the data were first screened for missing values and outliers. Skewness and kurtosis values were examined to assess normality, with all values falling within the acceptable range of −1 to +1, which indicated that the assumption of normality was met [54–56]. Confirmatory factor analysis and reliability analyses (Cronbach's Alpha ($\alpha$)) were conducted on the research scales. Pearson correlation analysis was performed to examine the bivariate relationships between the study variables. Finally, multiple regression analysis was performed, controlling for relevant variables, to investigate the predicted relationships between commitment to sports, satisfaction with sporting life, and mental and psychological well-being. Finally, multiple regression analysis was performed, controlling for relevant variables, to investigate the predicted relationships between commitment to sports, satisfaction with sporting life, and mental and psychological well-being. The basic assumptions of regression analysis, such as normality, linearity, homogeneity, independence, multicollinearity and absence of outliers, were examined. The confidence interval was evaluated as 95%. In addition to descriptive statistics, hypotheses were tested with correlation and multiple regression analyses within the scope of the relational model.

Confirmatory factor analysis is employed to verify the congruence between the predicted scale structure and the empirical data [46]. In this study, CFA was utilized to examine the model fit criteria. The analysis yielded the following fit indices: For the Sport Engagement Scale (SES), CMIN/DF($\chi^2$/df): 2.027, GFI:.973, AGFI:.954, CFI:.975, IFI:.975, TLI:.964, RMSEA:.047. The results of the CFA of the Sportive Life Satisfaction Scale (SLSS) were found as CMIN/DF($\chi^2$/df): 3.301, GFI:.973, AGFI:.954, CFI:.975, IFI:.975, TLI:.964, RMSEA:.070. The CFA results for the Mental Well-being Scale (WEMWBS) were CMIN/DF($\chi^2$/df): 2.285, GFI:.981, AGFI:.958, CFI:.985, IFI:.985, TLI:.974, and RMSEA:.052. For the Psychological Well-Being Scale (PWBS), the analysis yielded the following fit indices: CMIN/DF($\chi^2$/df): 2.819, GFI:.943, AGFI:.915, CFI:.958, IFI:.958, TLI:.945, and RMSEA:.062. The Cronbach's Alpha coefficients [57,58] (Table 2) and model fit indices were considered to fall within the ranges specified in the literature, indicating that the obtained values demonstrated excellent fit parameters and all scales were validated with the collected data [59–62].

 

## Result

Correlational analysis was conducted to examine the relationships between sport engagement, sportive life satisfaction, and well-being [45]. Following the guidelines suggested by Norusis [63] correlation coefficients (r) were interpreted as indicating weak $(0 < r \leq 0.3)$ moderate $(0.3 < r \leq 0.7)$ or strong $(0.7 < r \leq +1)$ relationships. The analysis revealed significant positive correlations between the subdimensions of the Sport Engagement Scale (SES) and the other well-being measures. Specifically, the *vigor* subdimension of the SES demonstrated moderate correlations with sportive life satisfaction $(r = .443)$, mental well-being $(r = .536)$, and psychological well-being $(r = .628)$. Similarly, the dedication subdimension of the SES showed moderate correlations with sportive life satisfaction $(r = .408)$, mental well-being $(r = .443)$, and psychological well-being $(r = .518)$. These findings indicate that higher levels of sport engagement, as reflected in both vigor and dedication, are associated with greater sportive life satisfaction, mental well-being, and psychological well-being (Table 3).

In order to assess the predictive power of sport engagement on sportive life satisfaction and well-being, multiple regression analysis was conducted. Prior to the analysis, multicollinearity was assessed using variance inflation factor (VIF) values. All VIF values were below 10 [64], indicating that multicollinearity was not a concern in the present data. The Durbin-Watson statistic was also examined to assess autocorrelation. The obtained value fell within the acceptable range of 1.5 to 2.5 [65], suggesting no significant autocorrelation. As noted by Tabachnick & Fidell [66], regression analysis allows for the prediction of one variable from another. In this case, sport engagement was used as the independent variable to predict the dependent variables: sportive life satisfaction, mental well-being, and psychological well-being. The results of this analysis are presented in Table 4.

The multiple regression model predicting sportive life satisfaction from sport engagement was statistically significant $(F_{(2-470)} = 67.010; p < 0.001)$. The model in the first part of Table 4 explained approximately 22% of the variance in sportive life satisfaction, as indicated by the $R^2$ value of .222 (adjusted $R^2 = .219$). Examination of the beta coefficients revealed that both subdimensions of sport engagement were significant positive predictors of sportive life satisfaction: Vigor: $\beta = .308$ indicated that higher levels of vigor were associated with greater sportive life satisfaction. Dedication: $\beta = .209$ suggested that increased focus also contributed to higher levels of sportive life satisfaction. Such findings seem to support the hypotheses $H_1$ and $H_2$, which proposed that both vigor and dedication, as components of sport engagement, would positively predict sportive life satisfaction.

**Table 2. Confirmatory factor analysis and validity & reliability analyses.**

| | CMIN/DF ($x^2$/df) | GFI | AGFI | CFI | IFI | TLI | RMSEA | Cronbach's Alpha (α) |
|---|---|---|---|---|---|---|---|---|
| Sport engagement | 2.027 | .973 | .954 | .975 | .975 | .964 | .047 | .83 |
| Sportive life satisfaction | 3.301 | .994 | .958 | .992 | .993 | .962 | .070 | .77 |
| Mental well-being | 2.285 | .981 | .958 | .985 | .985 | .974 | .052 | .91 |
| Psychological well-being | 2.819 | .943 | .915 | .958 | .958 | .945 | .062 | .86 |

**Table 3. Results of correlation analysis of research variables.**

| Variables | M | SD | 1 | 2 | 3 | 4 | 5 |
|---|---|---|---|---|---|---|---|
| Vigor | 6.42 | .562 | – | | | | |
| Dedication | 6.14 | .798 | .644** | – | | | |
| Sportive life satisfaction | 5.95 | .856 | .443** | .408** | – | | |
| Mental well-being | 4.18 | .575 | .536** | .443** | .535** | – | |
| Psychological well-being | 6.03 | .770 | .628** | .518** | .580** | .834** | – |

*\*\*p < 0.01; M: Mean; SD: Standard deviation; 1: vigor, 2: dedication, 3: sportive life satisfaction 4: mental well-being, 5: psychological well-being*

**Table 4. The impact of sport engagement on sportive life satisfaction, mental well-being and psychological well-being (multiple regression analysis results).**

**1- First Part**

| Model | B | Std. Error | Beta (β) | t | p | | |
|---|---|---|---|---|---|---|---|
| (Constant) | 1.564 | .400 | - | 3.912 | .000 | R=.471 R²=.222 Adj. R²= .219 | F(2-470)= 67.010 p=.000 D-W=1.870 VIF=1.710 |
| Vigor | .469 | .081 | .308 | 5.796*** | .000 | | |
| Dedication | .224 | .057 | .209 | 3.928*** | .000 | | |
| *Dependent variable: Sportive Life Satisfaction* | | | | | | | |
| 2- Second Part | | | | | | | |
| (Constant) | .633 | .254 | - | 2.488 | .013 | R=.551 R²=.303 Adj. R²= .300 | F(2-470)= 102.295 p=.000 D-W=1.976 VIF=1.710 |
| Vigor | .438 | .051 | .428 | 8.498*** | .000 | | |
| Dedication | .121 | .036 | .167 | 3.322** | .001 | | |
| *Dependent variable: Mental Well-being* | | | | | | | |
| 3- Third Part | | | | | | | |
| (Constant) | .460 | .311 | - | 1.477 | .140 | R=.645 R²=.416 Adj. R²= .414 | F(2-470)= 167.616 p=.000 D-W=2.065 VIF=1.710 |
| Vigor | .689 | .063 | .503 | 10.924*** | .000 | | |
| Dedication | .187 | .044 | .193 | 4.196*** | .000 | | |
| *Dependent variable: Psychological Well-being* | | | | | | | |

*Beta (β): Standardized coefficients; D-W: Durbin Watson; Variance Inflation Factor;**p<.01; ***p<.001.*

The model predicting mental well-being from sport engagement was statistically significant ($F_{(2-470)}$= 102.295; p<0.001). The model in the second part accounted for 30% of the variance in mental well-being, as indicated by the R² value of.303 (adjusted R²=.300). Examination of the beta coefficients showed that both subdimensions of sport engagement were significant positive predictors of mental well-being. Vigor: β=.428 demonstrated that higher levels of vigor were associated with greater mental well-being, while dedication: β=.167 showed that increased dedication also contributed to enhanced mental well-being. These results appear to support hypotheses $H_3$ and $H_4$.

The model presented in the third part of the table produced statistically significant results ($F_{(2-470)}$= 167.616; p<0.001). This model explained 41% of the variance in psychological well-being, as evidenced by the R² value of.416 (adjusted R²=.414). Analysis of the beta coefficients revealed that both subdimensions of sport engagement were significant positive predictors of psychological well-being. The first significant positive effect was in the sport engagement subdimension vigor (β=.503), and the second positive effect was in the dedication dimension (β=.193), which provided support for hypotheses $H_5$ and $H_6$. In summary, the regression analyses consistently demonstrated that sport engagement, specifically the vigor and dedication subdimensions, positively predicts sportive life satisfaction, mental well-being, and psychological well-being in athletes.

## Discussion

This study provides further evidence for the positive impact of sport engagement on various facets of well-being, including sportive life satisfaction. The observed positive and moderate correlation between sport engagement and sportive life satisfaction aligns with a growing body of literature highlighting the beneficial effects of sport participation on quality of life. The findings of the multiple regression analysis, demonstrating the positive predictive power of both vigor (approaching sport with energy and enthusiasm) and dedication (a strong commitment to their training and performance) on sportive life satisfaction, are consistent with previous research. For instance, Herbert et al. [67] presented evidence for the positive influence of sport participation on quality of life. Likewise, Martin et al. [68] reported high levels of life satisfaction among wheelchair basketball athletes, attributed to their engagement in sport. Zheng et al. [69] found a significant positive effect

of sport engagement on life satisfaction in athletes participating in the National Paralympic Games. Wu et al. [70] reported a positive relationship between quality of life and sport participation in young athletes. Gul & Kucukibis [71] concluded that sports high school students, with their higher levels of sport engagement, also experienced greater life satisfaction compared to other students. Beyond the athlete population, research has also linked commitment to physical activity with increased life satisfaction in university students [72]. Zhou et al. [73] demonstrated the indirect positive effects of physical exercise on life satisfaction in Chinese university students. Toros et al. [74] further emphasized this link, stating that individuals committed to regular physical activity report higher life satisfaction than sedentary individuals. These findings are further corroborated by numerous other studies [75,76; 77–80] that have collectively established a robust link between sport and physical activity engagement and enhanced quality of life and life satisfaction. The present study contributes to this body of knowledge by specifically highlighting the role of both vigor and dedication, as components of sport engagement, in promoting sportive life satisfaction among athletes. In this context, when an increase in an individual's level of engagement with sports is observed, it can be assumed that there may also be an increase in their sportive life satisfaction. Indeed, there can be many sources of motivation for individuals to participate in physical activities. In fact, this motivation may be related to many other psychological indicators. Therefore, it is thought that individuals who participate in physical activity with vigorus and dedicated motivation are also more likely to achieve satisfaction in sports.

The study further strengthens the evidence base for the positive relationship between sport engagement and mental well-being. The multiple regression analysis, which revealed that both vigor and dedication positively predict mental well-being, adds to the understanding of this relationship, which is consistent with Appelqvist-Schmidlechner et al. [81], who reported positive effects of sport participation on the mental well-being of young adult male athletes in Finland. Ochoa Del-Toro et al. [82] further corroborated this finding in a study with the Spanish population, indicating that sport engagement is associated with enhanced mental well-being across different populations. The benefits of sport engagement extend beyond athletes to individuals in various life contexts. For example, Guo and Jiang [83] found that physical activity positively affects the mental health of teachers, who often experience high levels of stress. Al-Johani [84] similarly reported a positive association between physical activity levels and mental well-being among teachers. Further supporting the link between sport engagement and mental well-being, Kim et al. [85] provided evidence that an optimal amount of physical activity (2.5 to 7.5 hours per week) has positive effects on mental health. In a Scottish study, Hamer et al. [86] also concluded that the intensity of participation in sporting activities was positively associated with mental health benefits. Such findings, along with the results of the present study, strongly suggest that sport engagement, involving both vigor and dedication, plays a crucial role in promoting mental well-being. Participation in physical activities may be related to the activation of individual self-control systems. Consistently following sports activities requires discipline. It is important to remember that the individual's mental health is also a significant factor in all these processes. Considering that mental states such as alertness and focus are important factors in sports engagement, it can be expected that individuals will also be sufficiently healthy mentally. Indeed, it is thought that individuals who show an increase in their level of engagement to sports may also experience an improvement in their mental health at the same time.

The final key finding of this study is the positive and moderate correlation between sport engagement and psychological well-being. The multiple regression analysis further revealed that both vigor and dedication, as components of sport engagement, have positive effects on psychological well-being, which aligns with a substantial body of research highlighting the benefits of physical activity and sport on various aspects of psychological health. D'Aurizio et al. [87] emphasized the importance of physical activity type, intensity, and frequency in protecting and enhancing psycho-physical well-being and cognitive functioning. Rodriguez-Bravo et al. [88] found that physical-sport activities positively influenced the psychological well-being of youth in both Spain and Colombia. Congsheng et al. [89] revealed similar positive effects of sport participation on the mental health of university students in Malaysia. Vella et al. [90] further confirmed the positive relationship between physical activity and mental health in their research. Edwards [91] reported significant improvements in various components of psychological well-being, including mood, life satisfaction, courage, and stress coping, following

a 2- to 6-month period of regular physical exercise. The positive association between physical activity and psychological well-being has been consistently reported in the literature [92]. Besides, studies by Pourranjbar et al. [93], Arruza et al. [94], and Donaldson & Ronan [95] have shown that increased sport engagement is linked to higher levels of psychological and behavioral well-being. In summary, the vigor and dedication involved in sport participation can be said to contribute to enhanced psychological health.

As highlighted by Ramazanoglu et al. [96], sport plays a vital role in physical, mental, and social development. Eime et al. [97] systematically reviewed the literature and found that sport participation offers numerous psychological and social advantages for children and adolescents, including increased self-esteem, enhanced social interaction, and reduced depressive symptoms. Miller & Hoffman [98] presented similar findings, underscoring the positive impact of sport on youth development. The inherent enjoyment associated with sport participation, as noted by Joung et al. [99], fosters continued engagement and creates an upward spiral of positive effects, contributing to increased social welfare [32]. The positive effects of exercise activities [100] will translate into increased feelings of satisfaction among individuals over time; because today's generations hate boredom and love when their parents initiate activities for them [101]. This highlights the potential for sport to promote individual well-being and contribute to broader societal benefits. In various social and cultural structures, factors such as socio-economic level, state and government policies, geographical conditions, traditions, etc. can have an impact on individuals' participation in physical activities. Indeed, in Türkiye, where the research was conducted, there are cultural, social, and political factors that influence individuals' commitment to sports. In particular, the state's reward system for successful athletes and the support mechanisms that will enable successful athletes to work at various levels of sports (coaches, educators, managers, etc.) in the future are in a position to encourage engagement to sports. Sporting activities, which have been disrupted in recent years due to the pandemic and natural disasters, have quickly entered a recovery phase as a result of the policies implemented. In this context, administrative processes have been successfully implemented to ensure the sustainability of all sporting activities. Therefore, the authorities have not overlooked the importance of utilizing the positive effects of participation in physical activities on individuals as a means to eliminate the negative experiences.

## Conclusions

The present study reinforces these notions by demonstrating the positive effects of sport engagement on sportive life satisfaction, mental well-being, and psychological well-being in athletes. These findings are particularly significant considering the detrimental consequences of low life satisfaction and poor mental health, which can include loneliness, stress, depression, and even suicidal tendencies [102–104]. Within the scope of the research, data were collected from athletes operating in various branches such as football, rugby, wrestling, kickboxing, archery, table tennis, taekwondo, volleyball, swimming, cycling, handball and karate. The diversity of sports disciplines is important in terms of reflecting the psychometric characteristics of different groups in the research. Likewise, the fact that the sample group is close to each other in terms of gender contributes to the research. It is thought that the fact that each sport branch has its own unique participant profile increases the possibility of generalizing the research results. The type of physical activity in which individuals with culturally distinct characteristics participate provides an opportunity to observe them psychologically. This study, which includes participants from diverse backgrounds such as different cultures, different sports disciplines, and different genders, is expected to provide insights across a wide range of perspectives based on its findings. Herefore, this study underscores the importance of encouraging and supporting sport participation across all age groups for improved well-being and a better quality of life. Furthermore, this research contributes to the existing literature both theoretically and practically. Existing literature highlights the effects of individuals participating in physical activity on general health indicators [2–5,6,7]. However, participation in physical activities and individuals' commitment to sports contain some differences. This study demonstrates that individuals who choose a physically active lifestyle are both physically and psychologically fit by placing physical activities at the center of their lives. Therefore, increasing social welfare will be possible with physically

and mentally resilient individuals. The contribution of the research results to the literature is important in that the relationship observed in the current health indicators of individuals committed to sports opens the door to multidisciplinary inferences. Theoretically, this study provides evidence that commitment to sporting activities positively influences both sport life satisfaction and mental well-being. This finding expands our understanding of the psychological benefits associated with sports participation. Practically, the positive correlation between sport engagement and well-being suggests that fostering a sense of dedication to sports may be a valuable strategy for enhancing individuals' mental health and quality of life.

## Limitations

While this study provides valuable insights into positive effects of sport engagement on life satisfaction, mental and psychological well-being, it is important to acknowledge its limitations. First, the study sample consisted solely of Turkish athletes, so the generalizability of the findings to athletes from other cultural backgrounds is limited. On the other hand, the fact that participants were included in the study using a convenience sampling method may limit generalizability due to the lack of randomization. In addition, the earthquake disaster in Turkey caused trauma on society. Due to the earthquake, sporting activities have been suspended for a period of time and athletes' routine physical activities have been disrupted. Therefore, delays in the data collection process due to the athletes being expected to return to their routine sports activities and move away from the negative effects of the earthquake can be considered among the limitations of the research. Secondly, our participants represented a limited number of sports and disciplines. Given the vast array of sports with their diverse demands and characteristics (seewikipedia.org/wiki/List_of_sports), further studies with athletes from a wider range of disciplines can contribute to the relevant literature. Finally, we relied on self-reported measures using Likert-type scales, which are subject to potential biases inherent in self-report data. Future research utilizing experimental or longitudinal designs could be more valuable in establishing the directionality of the observed relationships.

## Directions for further research and practical recommendations

The findings of this study have implications for various stakeholders, including athletes, families, coaches, educators, and policymakers. Future research could explore the specific mechanisms through which sport commitment enhances well-being, and investigate potential moderating factors such as age, gender, and type of sport.

Practically, this research underscores the importance of promoting sports participation and sports commitment. Policymakers could consider initiatives that increase access to sports facilities and programs. It may be recommended that bureaucrats and sports administrators discuss the possibility of enabling individuals to pursue both academic and athletic careers simultaneously. Additionally, for a society composed of physically and mentally healthy individuals, integrating physical activities into all levels of the education system can be proposed to the relevant authorities. On the other hand, it may be advisable to support initiatives undertaken through a public-private partnership model in rural and socio-economically disadvantaged regions with limited access to sports facilities and practices. Coaches and educators can play a crucial role by fostering a supportive and motivating environment that encourages athletes' dedication to their chosen sports. During the training of coaches, educators and managers, it can be suggested that in addition to physical capacity development programs, it may be useful to implement training programs such as sports psychology, sports sociology, sports philosophy, sports pedagogy, etc. Parents can also contribute by supporting their children's involvement in sports and promoting a positive attitude towards physical activity. In this context, it may be recommended that relevant institutions organize parent orientation training at regular periods to raise awareness among families. By encouraging widespread engagement in sports, society can reap the benefits of a physically and mentally healthier population. Increased sports participation can foster social interaction, enhance life satisfaction, and contribute to the development of productive and well-adjusted individuals. Future research should include more diverse samples to examine potential cross-cultural differences.

## Supporting information

**S1 Data. Data.**

(XLSX)

## Author contributions

**Conceptualization:** Arif Özsarı, Mehmet Çağrı Çetin, Mehmet Altin.

**Data curation:** Halil Uysal, Murat Tilki.

**Formal analysis:** Gültekin Lekesiz, Erkan Gülgösteren.

**Investigation:** Tolga Tek.

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
