## [Decision Letter · Decision Letter 0]

4 Nov 2025

Dear Dr. ÖZSARI,

Thank you for submitting your manuscript to PLOS ONE. After careful consideration, we feel that it has merit but does not fully meet PLOS ONE’s publication criteria as it currently stands. Therefore, we invite you to submit a revised version of the manuscript that addresses the points raised during the review process.

We look forward to receiving your revised manuscript.

Kind regards,

Mustafa Can KOC, PhD

Academic Editor

PLOS ONE

**Journal Requirements:**

2. We note that your Data Availability Statement is currently as follows:

“All relevant data are within the manuscript and its Supporting Information files.”

3. Please amend the manuscript submission data (via Edit Submission) to include author M.Çağrı ÇETİN.

4. We note you have included a table to which you do not refer in the text of your manuscript. Please ensure that you refer to Table 1 and 4 in your text; if accepted, production will need this reference to link the reader to the Table.

**Additional Editor Comments:**

Dear Author(s)

The reviewer has completed the report.

Please make the corrections carefully.

Kind regards

Reviewers' comments:

Reviewer's Responses to Questions

**Comments to the Author**

1. Is the manuscript technically sound, and do the data support the conclusions?

Reviewer #1: Yes

Reviewer #2: Yes

2. Has the statistical analysis been performed appropriately and rigorously?

Reviewer #1: Yes

Reviewer #2: Yes

3. Have the authors made all data underlying the findings in their manuscript fully available?

Reviewer #1: Yes

Reviewer #2: Yes

4. Is the manuscript presented in an intelligible fashion and written in standard English?

Reviewer #1: Yes

Reviewer #2: Yes

Reviewer #1: INTRODUCTION

Strengths

The introduction comprehensively and systematically addresses the relationships between physical activity, participation in sports, life satisfaction, and well-being. The logical progression from general health benefits to psychological dimensions is strong and supported by current sources. The author successfully integrates WHO data and theoretical frameworks to emphasize the global importance of the topic. Key concepts such as participation, life satisfaction, and psychological well-being are clearly defined. Overall, the text presents an academic style, extensive knowledge of the literature, and a strong theoretical foundation that supports the study's objectives.

Weaknesses

The introduction section, despite being comprehensive, is overly detailed in places, which detracts from the clarity of the main theme. In particular, the sections on the general health benefits of physical activity are lengthy, delaying the transition to the study's main focus: “the effects of sports participation on life satisfaction and psychological well-being.” Although the conceptual explanations are presented sequentially, the transitions between sections could be made more fluid. Furthermore, the original aspect of the research and the gap in the literature it aims to fill are not sufficiently clear, which weakens the scientific contribution of the study. The purpose of the research is conveyed in general terms in the last paragraph, but the rationale, importance, and expected contributions of this purpose should be emphasized more clearly. In terms of language, some expressions appear unnecessarily emotional or exaggerated, which partially diminishes the academic impartiality of the text. A simpler, more focused, and coherent narrative should be preferred.

METHOD

Strengths

The methodology section is clear, systematic, and structured in a manner appropriate to the research objectives. The correlational survey model used is an appropriate choice for examining the relationships between variables. The high number of participants (n=473) and the balanced gender distribution increase the representativeness of the study. The detailed presentation of the validity and reliability values of the measurement tools constitutes a strong methodological aspect. The high Cronbach's alpha coefficients for each scale demonstrate the reliability of the data quality. Furthermore, the detailed reporting of confirmatory factor analysis (CFA) results supports the structural validity of the scales. The clear indication of ethics committee approval and the principle of voluntary participation demonstrates that the research was conducted in accordance with ethical standards.

Weaknesses

Although the methodology section is generally adequate, it is open to improvement in some respects. First, the rationale for choosing the relational screening model and how this model aligns with the research hypotheses should be explained more clearly. The use of convenience sampling in sample selection may limit representativeness; the possible effects of this situation have not been discussed. The possible effects of the earthquake conditions experienced during the data collection process on the research process or participant profile have also not been evaluated. In the scales section, the cultural adaptation processes and validation methods of the measurement tools could be explained in more detail. In the statistical analysis section, it is not specified which variables were included in the multiple regression and how the model assumptions were tested. Furthermore, adding a general flow chart or systematic summary table of the research analysis process at the end of the methods section would enhance the readability and methodological transparency of the section.

RESULTS

Additionally, it would be more appropriate to include Table 1 and Table 2 in the methods section of the findings. (Table 1 in the research group and Table 2 in the section explaining the measurement tools)

It can be seen that the table descriptions are above the table. It is recommended that the descriptions be placed below the table.

DISCUSSION

Strengths

The discussion section is quite successful in integrating the research findings with the existing literature. The findings have been systematically compared with previous studies, and similarities have been strongly emphasized. The author has addressed the relationships between sports participation and life satisfaction, mental and psychological well-being from a multidimensional perspective; by discussing the separate effects of the sub-dimensions of “vigor” and “dedication,” he has made an original contribution to the literature. The discussion has produced results at both the theoretical and practical levels, offering applicable recommendations, particularly for sports policies, coaches, and educators. The conclusion and recommendations sections clearly outline the ways in which the research could contribute to social welfare and mental health in general. The chapter demonstrates a strong academic standard in terms of academic language, logical coherence, and source integration.

Weaknesses

Although the discussion section is comprehensive, it needs improvement in some respects. First, the discussion of the findings largely repeats previous studies, and the original contribution of the research should be emphasized more clearly. The text places excessive emphasis on causal inferences in places, but these results are limited to a correlational model; this should be clearly stated in the discussion. The contextual interpretation of the findings (e.g., sports culture in Turkey or post-pandemic sports participation dynamics) is lacking. Furthermore, the impact mechanisms of the “vigor” and “dedication” sub-dimensions could be discussed in a more analytical manner. The limitations section is brief; the data collection method, sample diversity, and the possible effects of cultural factors on the results should be addressed in greater depth. Although the recommendations presented in the final section are valuable, some are of a general nature; adding more concrete strategies for policymakers and practitioners would strengthen the text.

Reviewer #2: Technically well-designed data were collected accurately, and the results obtained from the data were generally found to support the main idea presented in the article. Moreover, this study adequately possesses the necessary scientific characteristics. It is observed that this is a review study and that the data collected were obtained through valid and reliable scales.

The statistical analyses conducted in the study are up-to-date. Primarily, descriptive statistics were used, followed by inferential statistics such as correlation and regression analyses. Furthermore, confirmatory factor analysis was employed to test the model, and model fit indices were reported. From this perspective, the statistical analyses performed are sufficient to obtain accurate and meaningful results.

The authors discussed all the data obtained in the article and presented their hypotheses one by one. Additionally, the collected data and results are adequate to clearly reveal the hypotheses. All essential sections required by the relevant journal were included in the article, and the necessary formatting guidelines were followed.

Finally, the writing language of the article is clear, well-organized, and the results are explained in a simple and understandable manner.

**Do you want your identity to be public for this peer review?** For information about this choice, including consent withdrawal, please see our Privacy Policy

Reviewer #1: No

Reviewer #2: **Yes: ** Dr. Kubilay OCAL

---

## [Author Response · Author response to Decision Letter 1]

5 Dec 2025

Dear Referee,

Thank you for your meticulous review.

We have made the corrections and submitted it as a file.

Regards

---

## [Editor Report · Decision Letter 1]

16 Dec 2025

The Impact of Sport Engagement on Life Satisfaction, Mental and Psychological Well-Being Among Athletes

PONE-D-25-54474R1

Dear Dr. Sari,

We’re pleased to inform you that your manuscript has been judged scientifically suitable for publication and will be formally accepted for publication once it meets all outstanding technical requirements.

Kind regards,

Mustafa Can KOC, PhD

Academic Editor

PLOS One
---

## [Editor Report · Acceptance letter]

PONE-D-25-54474R1

PLOS One

Dear Dr. ÖZSARI,

I'm pleased to inform you that your manuscript has been deemed suitable for publication in PLOS One. Congratulations! Your manuscript is now being handed over to our production team.

Kind regards,

on behalf of

Assoc.Prof. Mustafa Can KOC

Academic Editor

PLOS One